# Hydrogen-Rich Water (HRW) Reduces Fatty Acid-Induced Lipid Accumulation and Oxidative Stress Damage through Activating AMP-Activated Protein Kinase in HepG2 Cells

**DOI:** 10.3390/biomedicines12071444

**Published:** 2024-06-28

**Authors:** Sing-Hua Tsou, Sheng-Chieh Lin, Wei-Jen Chen, Hui-Chih Hung, Chun-Cheng Liao, Edy Kornelius, Chien-Ning Huang, Chih-Li Lin, Yi-Sun Yang

**Affiliations:** 1Department of Medical Research, Chung Shan Medical University Hospital, Taichung 402, Taiwan; zinminid@gmail.com; 2School of Medicine, Chung Shan Medical University, Taichung 402, Taiwan; phoenix33343@gmail.com (S.-C.L.); korn3lius82@gmail.com (E.K.); 3Department of Orthopaedics, Chung Shan Medical University Hospital, Taichung 402, Taiwan; 4Department of Biomedical Sciences, Chung Shan Medical University, Taichung 402, Taiwan; cwj519@csmu.edu.tw; 5Department of Life Sciences and Institute of Genomics and Bioinformatics, National Chung Hsing University, Taichung 402, Taiwan; hchung@dragon.nchu.edu.tw; 6Department of Family Medicine, Taichung Armed Forces General Hospital, Taichung 411, Taiwan; milkbottle97@yahoo.com.tw; 7School of Medicine, National Defense Medical Center, Taipei 114, Taiwan; 8Department of Internal Medicine, Division of Endocrinology and Metabolism, Chung Shan Medical University Hospital, Taichung 402, Taiwan; cshy049@csmu.edu.tw; 9Institute of Medicine, Chung Shan Medical University, Taichung 402, Taiwan

**Keywords:** metabolic dysfunction-associated steatotic liver disease (MASLD), free fatty acid (FFA), hydrogen-rich water (HRW), 5′-AMP-activated protein kinase (AMPK), oxidative stress

## Abstract

Metabolic dysfunction-associated steatotic liver disease (MASLD) is characterized by excessive fat accumulation in the liver. Intracellular oxidative stress induced by lipid accumulation leads to various hepatocellular injuries including fibrosis. However, no effective method for mitigating MASLD without substantial side effects currently exists. Molecular hydrogen (H_2_) has garnered attention due to its efficiency in neutralizing harmful reactive oxygen species (ROS) and its ability to penetrate cell membranes. Some clinical evidence suggests that H_2_ may alleviate fatty liver disease, but the precise molecular mechanisms, particularly the regulation of lipid droplet (LD) metabolism, remain unclear. This study utilized an in vitro model of hepatocyte lipid accumulation induced by free fatty acids (FFAs) to replicate MASLD in HepG2 cells. The results demonstrated a significant increase in LD accumulation due to elevated FFA levels. However, the addition of hydrogen-rich water (HRW) effectively reduced LD accumulation. HRW decreased the diameter of LDs and reduced lipid peroxidation and FFA-induced oxidative stress by activating the AMPK/Nrf2/HO-1 pathway. Overall, our findings suggest that HRW has potential as an adjunctive supplement in managing fatty liver disease by reducing LD accumulation and enhancing antioxidant pathways, presenting a novel strategy for impeding MASLD progression.

## 1. Introduction

Metabolic dysfunction-associated steatotic liver disease (MASLD), previously known as nonalcoholic fatty liver disease (NAFLD), is a common condition characterized by excessive fat accumulation in the liver [1]. Pathologically, MASLD is defined by the accumulation of 5–10% of body weight in liver fat or the identification of vacuolar degeneration in the cytoplasm of more than 10% of liver tissue sections [2]. With a prevalence of 4.6 times that of the general population, 57.5–74% of individuals with MASLD are obese. However, approximately 15–25% of individuals with MASLD are asymptomatic [3]. This asymptomatic nature suggests that excess fat alone in hepatocytes may not initially impair liver function. Nevertheless, fat accumulation can progress to nonalcoholic steatohepatitis (NASH) [1], which can advance to cirrhosis. Cirrhosis significantly hampers hepatic function and increases the risk of hepatocellular carcinoma [4]. Undiagnosed MASLD presents significant health hazards due to lipid accumulation in liver cells, even in the absence of symptoms. Therefore, early-stage strategies for regulating lipid levels or cellular damage could potentially impede disease progression.

Important molecular alterations occur as NASH emerges from MASLD, and the most notable alteration is increased hepatocyte metabolic stress due to lipid toxicity [5]. This stress primarily results from the accumulation of neutral lipids, mainly triglycerides from diet and lipogenesis, contained in lipid droplets (LDs). LDs are dynamic organelles that function as a critical component of cellular lipid metabolism and energy homeostasis by storing neutral lipids within a phospholipid monolayer. The accumulation of LDs in hepatocytes is a key feature of MASLD, providing insights into lipid metabolism dysregulation and its contribution to hepatic steatosis, inflammation, and progression to more severe liver diseases such as NASH [6]. Free fatty acids (FFAs) are fundamental constituents of LDs as they are building blocks in triglyceride synthesis. This suggests that LD number and size are directly influenced by the intracellular supply of FFAs [7]. In addition to promoting lipogenesis, FFAs hinder hepatocyte lipid export in the form of very-low-density lipoprotein (VLDL), indirectly exacerbating lipid accumulation [8]. FFAs also cause oxidative stress and inflammation, increasing the risk of metabolic dysfunction and liver damage. These conditions worsen reactive oxygen species (ROS) generation, fueling hepatic inflammation and MASLD progression [9]. An increased level of FFAs also leads to insulin resistance, exacerbating hepatic lipid accumulation and highlighting the crucial role FFAs play in the etiology of MASLD and NASH [10]. Thus, reducing inflammation, enhancing insulin sensitivity, and maintaining mitochondrial function may be achieved by identifying effective methods to reduce ROS and alleviate oxidative stress, thereby reducing liver lipid accumulation [11]. When these mechanisms work in concert, they can lessen the rate of lipid-induced metabolic stress-induced liver dysfunction progression from MASLD to NASH [12].

Molecular hydrogen (H_2_) has garnered significant interest in the biomedical field due to its selective antioxidant properties and ability to modulate signal transduction and gene expression [13]. Due to its small size and diffusibility, H_2_ can easily pass through cell membranes and enter subcellular spaces to provide protection. Its primary physiological benefits derive from its capacity to neutralize cytotoxic ROS, protecting cellular structures from oxidative stress damage [14]. For example, Li et al. found that injections of hydrogen-rich saline considerably lessen liver fibrosis by upregulating the heme oxygenase-1 (HO-1) signaling pathway, crucial in antioxidant and anti-inflammatory activity, indicating the therapeutic potential of H_2_ in fibrotic liver diseases [15]. Hydrogen gas inhalation for patients with fatty liver was investigated in a 2022 clinical trial, showing that regular inhalation enhanced immunological and oxidative markers in serum and liver fat content [16]. Recent research suggests molecular H_2_ may have notable liver-protective effects primarily due to its reduction in oxidative stress and inflammation, important factors in liver disease pathogenesis [17]. However, the molecular mechanism by which it acts remains unclear. Therefore, we employed an in vitro model of FFA-induced hepatocyte LD accumulation to simulate MASLD. Recent studies by Yin et al. have revealed that hydrogen-rich solutions, besides their intrinsic antioxidant capacity, exert protective effects via 5′-AMP-activated protein kinase (AMPK) [18]. Since AMPK regulates lipid metabolism and inhibits LD formation in liver cells, H_2_ likely alleviates fatty liver via this pathway. To further validate the underlying mechanism by which hydrogen-rich water (HRW) inhibits lipid accumulation, we simulated MALSD using an in vitro model of FFA-induced LD accumulation in HepG2 hepatic cells and investigated whether HRW treatment could mitigate cell damage and the potential molecular mechanisms involved.

## 2. Materials and Methods

### 2.1. Chemicals and Antibodies

Oil red-O, Nile Red, 2′,7′-dichlorodihydrofluorescein diacetate (H_2_-DCFDA), compound C, 3-(4,5-dimethylthiazol-2-yl)-2,5-diphenyltetrazolium bromide (MTT), and diamidino-2-phenylindole (DAPI) were supplied by Sigma (Munich, Germany). The AMPK (sc-74461), phospho-AMPK (sc-33524), Nrf2 (sc-722), HO-1 (sc-390991), CD36 (sc-7309), and p62 (sc-48402) antibodies were provided by Santa Cruz Biotechnology (Santa Cruz, CA, USA). The β-actin (NB600-501) and LC3B (NB100-2220) antibodies were provided by Novus Biologicals (Littleton, CO, USA), and the SOD1 (GTX100659), catalase (GTX110704), and Sirt1 (GTX61042) antibodies were obtained from GeneTex (Irvine, CA, USA). Primary antibodies were diluted at 1:200 (fICC) or 1:1000 (WB) in 0.1% Tween 20 blocking solution, while secondary antibodies were diluted at 1:500–1:5000. All chemicals were dissolved in phosphate-buffered saline (PBS) solution and stored at −20 °C before use in the experiments.

### 2.2. Generation of HRW and Determination of H_2_ Content and Oxidation–Reduction Potential (ORP)

HRW was produced by immersing a metallic magnesium stick (HD-50, Mg Chips 140 g, Unitiva Applied Materials Corp., Taipei, Taiwan) in distilled water at a flow rate of 200 mL/min. The stick, composed of 99.99% pure metallic magnesium, was stored in a polypropylene and ceramic container [19]. After purification, HRW was diluted to appropriate concentrations for experiments using a Millipore (Bedford, MA, USA) reverse osmosis water purification system. The medium powder was promptly dissolved in the newly acquired HRW. Following filtration, a dissolved hydrogen meter (ENH-1000, Trustlex Co., Ltd., Tokyo, Japan) was used to analyze H_2_ content and ORP values at various durations.

### 2.3. Cell Culture and Viability Assay

The human hepatoblastoma HepG2 cells were obtained from the American Type Culture Collection (Bethesda, MD, USA). The cells were cultured at 37 °C in 5% CO_2_ in Eagle’s Minimal Eagle’s Medium (EMEM) supplemented with 10% FBS, 100 units/mL penicillin, 100 µg/mL streptomycin, and 2 mM L-glutamine. The cells were used within 10 passages and checked for mycoplasma contamination every two months using the Mycoplasma PCR Detection Kit (Applied Biological Materials, Richmond, BC, Canada) according to the manufacturer’s instructions. HRW treatments were conducted at 37 °C. The culture medium was sterile-filtered and supplemented with freshly prepared HRW and 10% FBS, and the H_2_ concentration was adjusted to 300 ppb. For FFA treatment, palmitate and oleate were mixed in a 1:2 ratio by conjugation with bovine serum albumin (BSA) to simulate natural fatty acid conditions. Specifically, palmitic acid and oleic acid were dissolved in ethanol and heated gently at 70 °C with continuous stirring, and sodium hydroxide was added dropwise to ensure complete dissolution. The fatty acids were then conjugated with a 10% BSA solution, adjusted to pH 7.4, mixed thoroughly, incubated at 37 °C for 1 h, and filter-sterilized, resulting in a final concentration of 1 mM for treatments [20]. The vehicle used was a BSA solution without FFAs, and the treatment with FFAs or vehicle was conducted for 24 h. For the viability assay, the cells were seeded into 24-well plates at a density of 1 × 10^5^ cells/well and treated as instructed, and MTT was added to assess cell viability after 24 h of treatment. Live cells convert MTT to a purple formazan product, detected spectrophotometrically at 550 nm. Viability was determined by calculating the percentage of control cells that received only vehicle treatment.

### 2.4. Lipid Accumulation Assay

HepG2 cells were grown to an initial density of 1 × 10^5^ cells/well in 24-well plates and exposed to 1 mM FFAs for 24 h. The cells underwent three cycles of ice-cold PBS washing before being fixed for 30 min with 4% paraformaldehyde. After fixation, the cells were washed with PBS and stained for 15 min at room temperature using an Oil red-O working solution (0.5 g of Oil red-O powder dissolved in 60% ethanol). PBS washing removed the unbound stain. Dimethyl sulfoxide (DMSO) was added to each sample and shaken at room temperature for five minutes, and the absorbance at 510 nm was measured to determine the Oil red-O content. DMSO primarily serves to dissolve the Oil red-O dye from the cells, enabling subsequent spectrophotometric quantification to measure the lipid content accurately.

### 2.5. Cholesterol and Triglyceride Measurements

After incubation, the cells were resuspended in 1 mL of PBS following two PBS washes (1 × 10^6^ cells/mL). The 0.3 mL cell suspension was sonicated for two minutes, and 0.1 mL of the cell extract was used to measure triglycerides or cholesterol. Total cholesterol and triglycerides were measured using an enzyme-based method with the CheKine colorimetric assay kits (Abbkine, Atlanta, GA, USA). Protein content was determined using the BCA protein assay kit (Millipore). All procedures were performed according to the manufacturer’s guidelines.

### 2.6. Nile Red Staining and High-Content Analysis

After treatment, cells were fixed for 15 min with 4% paraformaldehyde and cleaned with PBS. Nile Red (1 μg/mL) staining was performed at room temperature for five minutes in the dark. Imaging was conducted using an ImageXpress Micro Confocal High-Content imaging system (Molecular Devices, Sunnyvale, CA, USA). With at least 200 cells viewed per well and each condition run in triplicate, quantitative analysis of Nile Red-stained particles was automated using the custom module in the MetaXpress software (Version 6.5.3.427, Molecular Devices).

### 2.7. Reactive Oxygen Species (ROS) Measurement

The cell-permeant H_2_-DCFDA method was used to measure intracellular ROS levels. After the cells were exposed to 10 µM H_2_-DCFDA for 30 min, a flow cytometer (FACSCanto II, BD Biosciences, Bedford, MA, USA) was used to quantify the oxidative burst. Data were analyzed using BD FACSDiva software version 5.0.

### 2.8. Mitochondrial Membrane Potential Analysis

The cationic dye JC-1, which accumulates in mitochondria in a potential-dependent manner, was used to evaluate mitochondrial function. Normal cells contain a red fluorescent JC-1 monomer; under stress, JC-1 aggregates and emits green fluorescence. Following treatment, the cells were incubated with 1 µM JC-1 for 30 min at 37 °C. After the removal and cleaning of the staining medium, imaging was conducted using an inverted fluorescence microscope (DP72/CKX41, Olympus, Tokyo, Japan). The red/green fluorescence ratio was quantitatively examined using Image J software (Version 1.54j, National Institutes of Health, Bethesda, MD, USA).

### 2.9. Lipid Peroxidation Measurement

Following treatment, the cells were collected by centrifuging them for 10 min at 3000 rpm and discarding the supernatant. The cells were resuspended in 300 µL of 10% medium and sonicated for one minute on ice. After adding 100 µL of precooled PBS and removing the culture solution, the cells were sonicated and kept at −20 °C. Malondialdehyde (MDA) content was assessed using the Lipid Peroxidation Assay Kit (BioVision, San Francisco, CA, USA) by colorimetric mode (OD = 532 nm), and protein concentration was determined using a BCA protein assay kit (Millipore). MDA content was expressed as nmol/mg protein.

### 2.10. Western Blot Analysis

Gold Lysis Buffer was used to prepare and quantify cell lysates. Equal protein amounts were separated using SDS-PAGE and transferred to a PVDF membrane. After blocking, the membranes were probed with primary and HRP-conjugated secondary antibodies. Amersham ECL detection reagents were used to visualize protein signals, and an AI600 Imaging System (GE Healthcare, Chicago, IL, USA) was used to capture chemiluminescence images from the membranes.

### 2.11. mRNA Expression Analysis by Reverse-Transcription Quantitative PCR (qRT-PCR)

Total mRNA was extracted and quantified using the RNeasy Kit (Qiagen, Germantown, MD, USA). mRNA was reverse transcribed to cDNA using specific conditions with a TProfessional Thermocycler (Biometra, Göttingen, Germany). qRT-PCR was performed using Power SYBR Green PCR Master Mix and the ABI 7300 Sequence Detection System (Applied Biosystems, Foster City, CA, USA). The following cycling conditions were applied: initial denaturation at 95 °C for 10 min, 40 cycles of 95 °C for 15 s, 60 °C for 1 min, followed by a dissociation step. Relative mRNA expression levels were calculated and normalized to GAPDH expression using Sequence Detection Systems software (v2.4.1, Applied Biosystems). Each primer pair utilized was developed with Primer Express software (v3.0, Applied Biosystems), as indicated in Table 1.

### 2.12. Statistical Analysis

The data are presented as mean ± standard error of the mean (SEM). Analysis of variance (ANOVA) was used in the statistical analyses, and Dunnett’s post hoc test was used for multiple comparisons. Using SPSS Statistics 25 software (SPSS, Inc., Chicago, IL, USA), the statistical significance was set at *p* < 0.05 and *p* < 0.01. We conducted all relevant experiments in independent triplicates.

## 3. Results

### 3.1. Attenuation of High-FFA-Induced Lipid Accumulation in HepG2 Cells by HRW

Initially, we investigated the changes in hydrogen molecule content over time in the culture medium solution generated using HRW, in the absence of FBS and cells. As depicted in Figure 1A, the H_2_ concentration of freshly prepared culture medium reached about 400 ppb. However, over time, the concentration of H_2_ in the solution gradually diminished, maintaining a level of about 100 ppb after 24 h. To further confirm that H_2_ in these solutions retained oxidation-reduction capacity, we measured changes in the ORP over 24 h. As shown in Figure 1B, the redox potential of HRW increased from −200 mV when freshly prepared but remained below −50 mV within 24 h, indicating that the HRW culture medium retained a certain reduction capacity during this period. Subsequently, we assessed the effects of HRW on HepG2 cells. The results demonstrated no significant differences in cell viability and growth rate between HRW-treated cells and those in a normal culture medium (Figure 1C). The 24 h timeframe for HRW studies was selected to capture the short-term effects of HRW on lipid accumulation and oxidative stress, based on previous studies suggesting significant changes within this period [21]. Therefore, we employed Oil red-O staining to detect lipid content in HepG2 cells and evaluate the impact of HRW on intracellular lipid accumulation. As shown in Figure 1D, compared to the control group, Oil red-O staining revealed a marked increase in lipid accumulation in HepG2 cells treated with a high level of FFAs (1 mM) over 24 h. In contrast, HRW appeared to reduce intracellular lipid accumulation induced by FFAs. Oil red-O was dissolved in dimethyl sulfide to quantify lipid content, and absorption was measured at 510 nm by spectrophotometry (Figure 1E). The results showed that lipid accumulation increased approximately five-fold, while HRW significantly inhibited this increase, suggesting that HRW effectively reduces high lipid accumulation induced by FFAs in HepG2 cells.

### 3.2. Reduction in LD Size by HRW in FFA-Induced HepG2 Cells

To further confirm the inhibition of intracellular lipid accumulation, we analyzed changes in total lipid content. As shown in Figure 2A, compared to control groups, high FFAs significantly induced total lipid accumulation in the cells, but HRW markedly attenuated this effect. The analysis of neutral lipid triglycerides yielded similar results, demonstrating that HRW effectively reduces FFA-induced triglyceride accumulation in HepG2 cells. Previous studies have indicated that triglycerides are the main components of intracellular LDs and affect the physiological functions of hepatocytes [22]. As a result, we used Nile Red fluorescent dye to stain the cells and evaluated the effect of HRW on LDs. As depicted in Figure 2B, numerous LDs accumulated in HepG2 cells following a 24 h treatment with high FFAs. Conversely, co-treatment with HRW significantly reduced LD accumulation. To better understand the effect of HRW, we evaluated changes in parameters, including the number and size of LDs, using high-content analysis (HCA). As illustrated in Figure 2C, the analysis demonstrated that FFAs substantially increased the mean count of LDs exhibiting fluorescent signals within each cell compared to the control group. Nevertheless, the average number of LDs was not significantly impacted by HRW. Further categorization of the LDs according to their diameter into three groups—less than 1 μm, between 1 μm and 3 μm, and greater than 3 μm—revealed that while HRW does not affect the quantity of LDs, it does induce a tendency for the larger ones to diminish (Figure 2D). Collectively, HRW reduced the increase in high-FFA-induced LDs in HepG2 hepatocytes, which is considered detrimental to normal cell physiological function.

### 3.3. HRW Attenuates FFA-Induced Oxidative Stress and Reduces Lipid Peroxidation in HepG2 Cells

Previous studies have shown that excessive LD accumulation can increase oxidative stress in cells and cause cellular damage. Consequently, we examined whether HRW affected changes in intracellular oxidative stress. As shown in Figure 3A, diacetyl dichlorofluorescein (H_2_-DCFDA) staining indicated that the control and HRW-treated cells displayed baseline ROS levels, while FFA treatment alone increased ROS levels and slightly altered nuclear morphology, suggesting cellular stress. On the contrary, the combination of HRW and FFAs resulted in a noticeable decrease in ROS levels, indicating a synergistic effect that reduces cell stress or damage. To further quantify the changes in ROS levels, we analyzed H_2_-DCFDA staining results using flow cytometry. As shown in Figure 3B, compared to the control group, the intracellular ROS content was significantly elevated by FFA treatment alone, reaching a level that was nearly three times higher. Co-treatment with HRW significantly reduced ROS accumulation. Since mitochondria are one of the major sources of ROS in cells, we assessed mitochondrial function by measuring membrane potential using JC-1 staining. The results shown in Figure 3C showed that HRW ameliorated the decrease in mitochondrial membrane potential induced by FFAs, thus improving mitochondrial dysfunction. Subsequently, we investigated whether excessive ROS affected lipid peroxidation by analyzing malondialdehyde (MDA), a marker of oxidative damage to lipids. The results showed that FFAs significantly increased intracellular MDA content, whereas HRW significantly reduced the FFA-induced increase in MDA (Figure 3D), suggesting that HRW reduces lipid peroxidation.

### 3.4. AMPK Is Essential for Protective Capabilities of HRW

To elucidate the mechanism by which HRW reduces oxidative stress damage caused by FFAs in liver cells, we measured the expression changes of various genes related to antioxidant stress damage. As shown in Figure 4A, Western blot analysis revealed notably upregulated expression levels of antioxidant enzymes superoxide dismutase 1 (SOD1) and catalase by HRW, suggesting the presence of a cellular response counteracting increased ROS production compared to the FFA-treated group. Additionally, HRW induced the expression of nuclear factor erythroid 2-related factor 2 (Nrf2) and heme oxygenase (HO-1), indicating increased antioxidant enzyme levels. Treatment with the AMPK inhibitor compound C (CC) significantly reduced HRW’s induction effect, suggesting that HRW may counteract FFA-induced oxidative stress damage through the AMPK/Nrf2/HO-1 signaling pathway. Previous studies have indicated that activation of AMPK reduces fibrosis in liver cells [23]. Therefore, we explored whether HRW reduces the expression of genes related to hepatocellular fibrosis induced by excessive FFAs. As shown in Figure 4B, the qRT-PCR results demonstrated that fibrosis markers such as α-smooth muscle actin (α-SMA) and collagen type I were markedly lower in the HRW co-treated group compared to the FFA-induced group. Additionally, inflammation-related factors, including IL-1β, IL-6, and TNFα, were inhibited in the HRW group, indicating that HRW slows down inflammation and fibrosis in hepatic cells induced by excessive FFAs. Finally, it is an established fact that autophagy, including lipophagy, is responsible for the regulation of LD degradation. Experimental analysis of autophagy markers, including LC3-II and p62, was used to infer changes in lipophagy. This is because the reduction in the size of LDs is enhanced by increases in autophagy. As illustrated in Figure 4C, HRW restored the level of LC3-II expression, which had been inhibited by FFAs, and decreased the expression of p62. This indicated an increase in autophagy activity, as p62 was degraded during the autophagic process. Conversely, the autophagy promoted by HRW was diminished when compound C (CC) was administered concurrently, indicating that the activation of AMPK was also a significant factor in the influence of HRW on the reduction in LD particle size. The expression of CD36 did not appear to be significantly affected by HRW, despite the fact that CD36 was essential for the formation of LDs by facilitating the uptake and storage of FFAs.

## 4. Discussion

Recent studies have revealed significant interest in the connection between molecular hydrogen and its potential therapeutic impact on fatty liver disease [24]. Specifically, molecular hydrogen is a small, neutral molecule that is readily absorbed through cell membranes and has been shown to have numerous positive biological effects. Numerous clinical trials have been conducted to investigate the potential therapeutic benefits of hydrogen on major illnesses, including cancer, heart disease, and respiratory issues [25]. These investigations consistently demonstrated that molecular hydrogen has beneficial effects, particularly in reducing oxidative stress. Due to these properties, molecular hydrogen has been recognized for its potential to treat a range of metabolic diseases, including fatty liver disease [26]. However, the complete molecular mechanisms underlying the action of molecular hydrogen remain unclear. In this work, we administered molecular hydrogen in the form of HRW to HepG2 hepatocytes. Currently, the most prevalent methods of administering H_2_ include the direct inhalation of hydrogen or hydrogen sulfide gas, as well as the HRW used in our study. Considering factors such as patient acceptance, safety, and the convenience of raw material storage, we believe that HRW represents the most well-rounded solution at this time. Compared to the traditional direct administration of hydrogen gas, this mode of delivering H_2_ significantly enhances the safety of the administration process. Additionally, we were able to demonstrate the potential molecular mechanism of HRW’s protective effect against the accumulation of excess lipid LDs in hepatocytes. Due to the complex interplay of factors contributing to the progression of the disease and the lack of approved specific therapies, the treatment of NASH remains very challenging. Therefore, our findings indicate that molecular hydrogen demonstrates positive effects, particularly in reducing oxidative stress, an important factor associated with the progression of fatty liver to NASH, suggesting its promising role in treating fatty liver disease [27].

Disrupted lipid metabolism causes LDs to accumulate in hepatocytes in fatty liver disease. This can result in hepatic inflammation, steatosis, and ultimately more severe liver damage, such as steatohepatitis [22]. The enzyme AMPK plays a pivotal role in this context. It is well recognized that AMPK is an essential regulatory enzyme that supports cellular energy homeostasis and is activated by energy stress, generally experienced during cellular stress or energy demand [28]. Among the variables influencing the dynamics of LDs in hepatocytes, the control of lipid metabolism is significant [29]. By regulating important enzymes in this pathway through phosphorylation, such as sterol regulatory element-binding protein 1c (SREBP-1c), AMPK can inhibit lipogenesis [30]. LDs are primarily composed of fatty acids, and one of the main transcription factors regulating the expression of the enzymes involved in this process is SREBP-1c. Additionally, AMPK can stimulate lipolysis through the phosphorylation of lipases or coactivators involved in lipolysis, such as comparative gene identification-58 (CGI-58), facilitating lipid breakdown within LDs [31]. The hydrolysis of triglycerides stored in LDs is thereby increased. The evidence suggests that AMPK modulates LD size by controlling the expression and function of proteins implicated in the formation and maintenance of LDs. This is consistent with our findings, indicating that HRW primarily stimulates AMPK activity to achieve its effect on LD regulation. It is widely recognized that AMPK induces autophagy, including lipophagy, the process that regulates LDs. Our results demonstrate that HRW enhances cellular lipid metabolism and facilitates the degradation of LDs, resulting in a reduction in their size. Several studies suggest that hydrogen might directly impact the signaling pathways that activate AMPK. For example, it has been demonstrated that hydrogen influences upstream kinases that activate AMPK, such as liver kinase B1 (LKB1) and calcium/calmodulin-dependent protein kinase kinase 2 (CaMKK2) [32,33]. Hydrogen may facilitate the activation of AMPK and enhance its role in regulating cellular energy by modulating these kinases. As we previously reported, HRW can also alleviate neuronal stress by activating the Sirt1 and AMPK signaling pathways. [19]. These findings suggest that AMPK activation plays a significant mediating role in the function of molecular hydrogen, thereby contributing to its protective effect on tissues and cells.

HRW has also been shown to exert significant effects against oxidative stress damage. This role can be attributed to several mechanisms. Firstly, one of the main features that distinguish molecular hydrogen from other antioxidants is its capacity to specifically neutralize cytotoxic oxygen radicals. Hydrogen interacts exclusively with the most hazardous free radicals, including hydroxyl radicals and peroxynitrite, without affecting physiologically beneficial free radicals or molecules. This selectivity allows hydrogen to effectively prevent oxidative damage while preserving cellular signaling pathways essential for cell survival and growth [26]. Hydrogen also protects against oxidative stress by modulating cellular signaling pathways that strengthen endogenous antioxidant defenses. The Nrf2 pathway, a crucial regulator of cellular resistance to oxidants, has been demonstrated to be activated by hydrogen [34]. Activation of Nrf2 leads to the increased expression of numerous antioxidant and cytoprotective genes, such as those encoding superoxide dismutases and catalase, thereby enhancing resistance to oxidative stress and protecting hepatocytes from damage [35]. Additionally, molecular hydrogen positively impacts mitochondrial function, which is crucial for cellular energy production and health. Improved mitochondrial function is associated with reduced production of harmful ROS and enhanced ATP production, further mitigating oxidative stress and promoting cellular health [36]. Given that steatosis is a critical stage in the progression of fatty liver disease to more severe NASH, the accumulation of excessive LDs and a high oxidative stress environment can exacerbate disease progression. This underscores the potential of HRW to slow the development of fatty liver disease.

Collectively, this research represents the initial demonstration that HRW has the capability to regulate the dimensions of hepatocyte LDs induced by an excess of FFAs, thereby mitigating potential harm. Unlike studies utilizing a single type of FFA, the use of mixed FFAs in this study may more accurately represent physiological conditions. However, our findings are based on an in vitro investigation involving a single cell line and did not examine other cell types, such as stellate cells, which contribute more significantly to extracellular matrix (ECM) deposition. Relying solely on a single-cell model may limit the power of inferences from our experimental results. Therefore, further comprehensive animal experiments are necessary to determine the function and impact of HRW in the human body. Overall, our research indicates that molecular hydrogen is a potentially effective adjunctive therapy for managing fatty liver disease, highlighting its ability to slow the disease’s progression through antioxidant and anti-inflammatory actions. Further investigation is needed to fully determine its role and efficacy in clinical settings, particularly through extensive clinical trials.

## Figures and Tables

**Figure 1 biomedicines-12-01444-f001:**
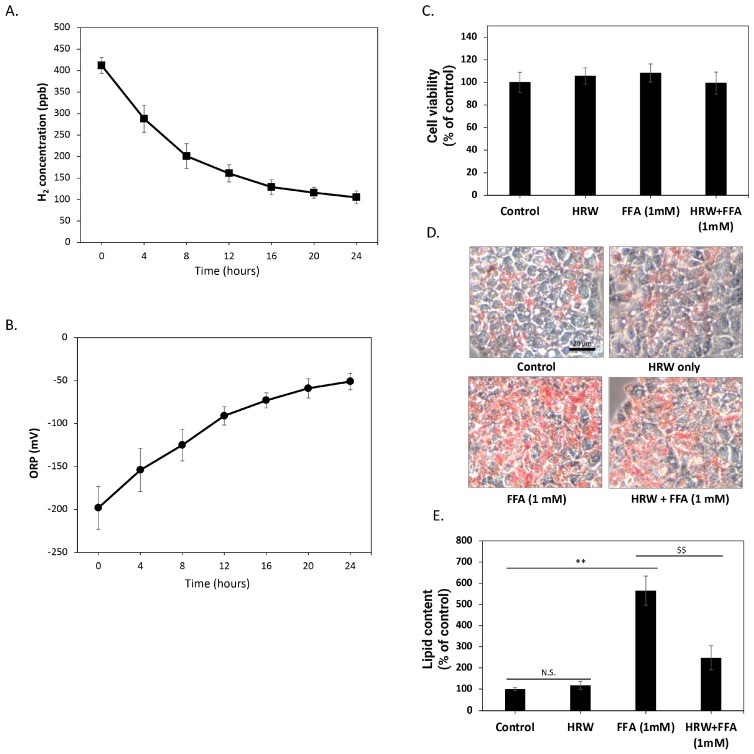
HRW reduced the high lipid accumulation induced by FFAs in HepG2 cells. (**A**) The H_2_ concentration analysis showed that the freshly prepared culture medium reached approximately 400 ppb. Subsequently, H_2_ concentrations gradually decreased but remained above 100 ppb after 24 h. (**B**) The oxidation-reduction potential (ORP) of HRW was analyzed, revealing that its value gradually increased from −200 mV to −50 mV over 24 h. (**C**) Cell viability was determined using MTT assays. Comparing HRW to the normal culture medium, the results revealed no significant differences in cell viability. (**D**) Microscopic observations of Oil red-O-stained lipids showed that HRW markedly reduced the accumulation of high-FFA-induced lipids after 24 h of treatment in HepG2 cells. Scale bar = 20 μm. (**E**) Quantitative analysis of Oil red-O staining was performed by measuring absorbance at 510 nm. Values are presented as the mean ± SEM of at least 3 different experiments. Nonsignificant *p*-values are indicated by N.S. An asterisk (*) indicates a significant difference compared to the control group (** *p* < 0.01), and a dollar sign ($) indicates a significant difference compared to the FFA-treated group ($$ *p* < 0.01).

**Figure 2 biomedicines-12-01444-f002:**
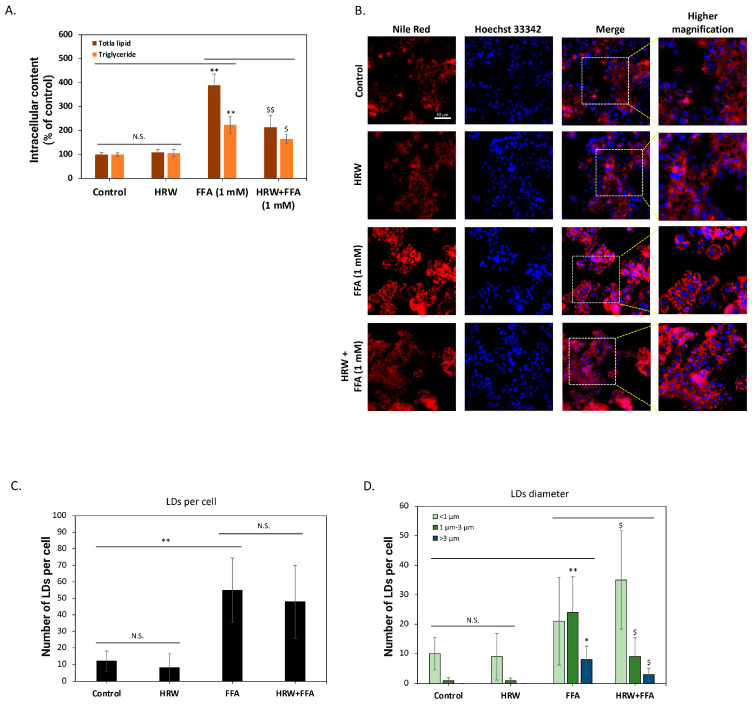
HRW reduced high-FFA-induced lipid drops (LDs) in HepG2 cells. (**A**) The determination of total lipids and triglycerides using enzymatic methods showed that HRW effectively reduced the accumulation of FFA-induced total lipids and triglycerides in HepG2 cells. (**B**) Fluorescent staining with Nile Red revealed that FFAs caused a significant accumulation of intracellular lipid drops (LDs), whereas the combination with HRW markedly reduced the accumulation of LDs. Scale bar = 50 μm. (**C**,**D**) High-content analysis (HCA) of fluorescence images indicated that although HRW did not effectively suppress the number of LDs caused by FFAs, it reduced the number of larger-diameter LDs. All data were collected from at least three independent experiments and are presented as mean ± SEM. Nonsignificant *p*-values are indicated by N.S. An asterisk (*) indicates a significant difference compared with the control group (* *p* < 0.05 and ** *p* < 0.01), and a dollar sign ($) indicates a significant difference compared with the FFA-treated group ($ *p* < 0.05 and $$ *p* < 0.01).

**Figure 3 biomedicines-12-01444-f003:**
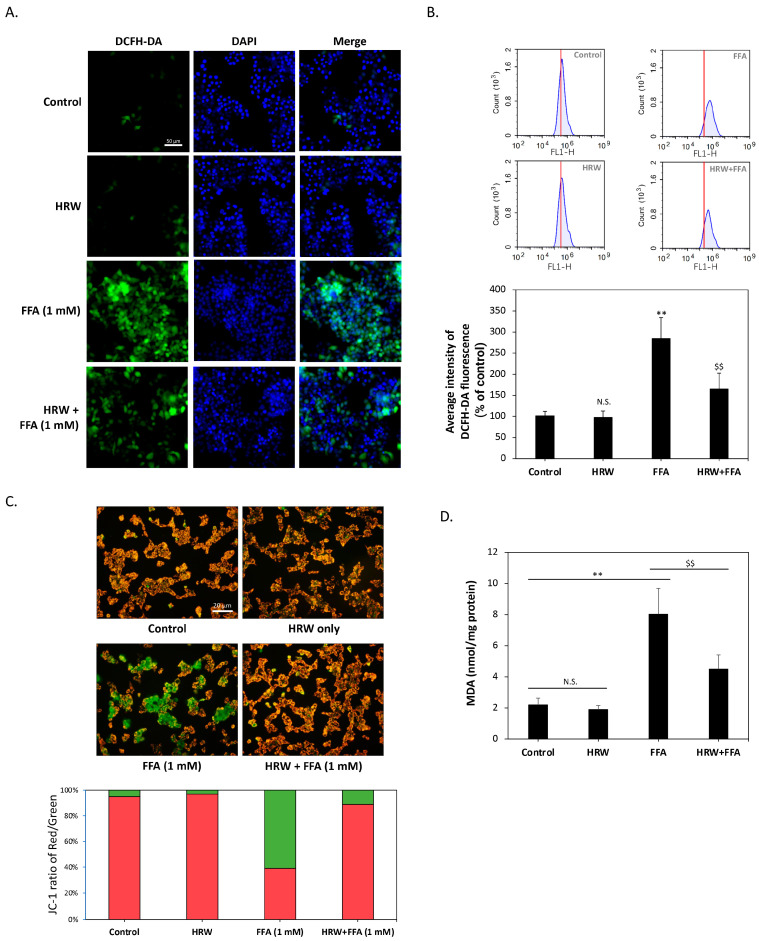
HRW lowered lipid peroxidation and FFA-induced oxidative stress. (**A**) Reactive oxygen species (ROS) levels in HepG2 cells were determined by H_2_-DCFDA staining. Scale bar = 50 μm. (**B**) Flow cytometry histograms showed that FFAs significantly increased ROS levels; however, HRW markedly reduced the amount of ROS induced by FFAs. (**C**) Mitochondrial membrane potential was measured using JC1 staining. JC1 aggregates appeared red, while monomers appeared green. HRW restored mitochondrial function by mitigating the decrease in membrane potential caused by FFAs. Scale bar = 20 μm. (**D**) Malondialdehyde (MDA) lipid peroxidation was measured in FFA and HRW treatments. HRW significantly attenuated the increase in MDA induced by FFAs. All data were collected from at least three independent experiments and are presented as mean ± SEM. Nonsignificant *p*-values are indicated by N.S. An asterisk (*) indicates a significant difference compared with the control group (** *p* < 0.01), and a dollar sign ($) indicates a significant difference compared with the FFA-treated group ($$ *p* < 0.01).

**Figure 4 biomedicines-12-01444-f004:**
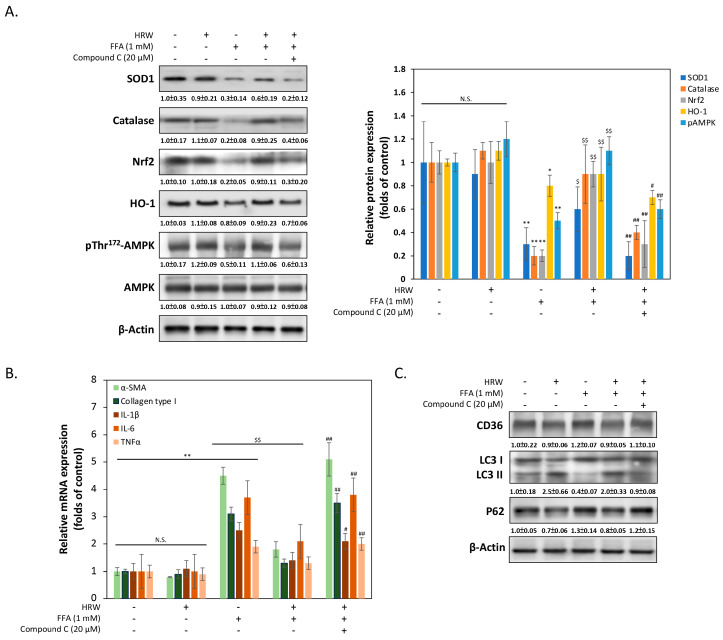
The AMPK pathway was stimulated by HRW to reduce FFA-induced oxidative stress. (**A**) The levels of SOD1, catalase, Nrf2, HO-1, pThr^172^AMPK, and AMPK were analyzed, with densitometric analysis of all Western blot bands normalized to β-actin. (**B**) The expression levels of several mRNAs involved in fibrosis and inflammation were compared using quantitative real-time PCR (qRT-PCR). (**C**) Representative Western blotting analysis of CD36, LC3 I/II, and p62 expression indicated that autophagy inhibited by FFAs was restored by HRW. However, the protective effect of HRW was downregulated by the AMPK inhibitor compound C, suggesting that HRW may trigger autophagy via AMPK. All data were collected from at least three independent experiments and are presented as mean ± SEM. Nonsignificant *p*-values are indicated by N.S. An asterisk (*) indicates a significant difference compared with the control group (* *p* < 0.05 and ** *p* < 0.01). A dollar sign ($) indicates a significant difference compared with the FFA-treated group ($$ *p* < 0.01). A hash symbol (#) indicates a significant difference compared with the FFA + HRW group (# *p* < 0.05 and ## *p* < 0.01).

**Table 1 biomedicines-12-01444-t001:** Primer sequence of different genes for qRT-PCR analysis.

Genes	Forward (5′-3′)	Reverse (5′-3′)
αSMA	TGCTCCAGCTATGTGTGAAGA	AGGTCGGATGCTCCTCTG
Collagen I	TGAGCCAGCAGATTGAGAACA	GGGTCGATCCAGTACTCTCCG
IL-1β	CACCTCTCAAGCAGAGCACAG	GGGTTCCATGGTGAAGTCAAC
IL-6	TCTGGAGTTCCGTTTCTACCTGG	CATAGCACACTAGGTTTGCCGAG
TNFα	AAATGGGCTCCCTCTCATCAGTTC	TCTGCTTGGTGGTTT GCTACGAC
GAPDH	TGGTATCGTGGAAGGACTCATGAC	ATGCCAGTGAGCTTCCCGTTCAGC

## Data Availability

The data that support the findings of this study are available from the corresponding author, Lin C.-L., upon reasonable request due to privacy or ethical restrictions.

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
