# Peer review of "Hydrogen-Rich Water (HRW) Reduces Fatty Acid-Induced Lipid Accumulation and Oxidative Stress Damage through Activating AMP-Activated Protein Kinase in HepG2 Cells"

_biomedicines, 2024, doi:10.3390/biomedicines12071444_

Round 1
Reviewer 1 Report
Comments and Suggestions for Authors
In this study, the authors investigated the accumulation of intracellular lipid droplets in hepatocytes induced by free fatty acids to better understand Metabolic Dysfunction-Associated Steatotic Liver Disease (MASLD). They used HepG2 cells as an in vitro model. Overall, the authors provided a well-designed manuscript with a thorough methods section. However, the biological significance and rationale of the study need further elaboration, and the primary objectives require more detailed clarification.
The introduction section effectively presents major topics, but my major concern is that the study solely relied on the human hepatoblastoma HepG2 cell line. Including additional cell models would enhance the study's robustness. The authors should provide a justification for their choice of this particular cell model. Additionally, it is important to specify whether cell viability was assessed immediately or after 24 hours, and explain why hydrogen-rich water (HRW) studies were performed only for 24 hours.
Certain claims in the manuscript are overly assertive and need to be revised. For instance, the statement that "molecular hydrogen has a promising role in treating fatty liver and consistently demonstrates positive effects, particularly in reducing oxidative stress, which is the main factor causing fatty liver to progress to NASH" should be rephrased. It would be more appropriate to say, "Molecular hydrogen demonstrates positive effects, particularly in reducing oxidative stress, an important factor associated with the progression of fatty liver to NASH, suggesting its promising role in treating fatty liver disease." Additionally, the study's limitations, such as the reliance on a single cell model, should be discussed in the discussion section.
Comments on the Quality of English LanguageIn this study, the authors investigated the accumulation of intracellular lipid droplets in hepatocytes induced by free fatty acids to better understand Metabolic Dysfunction-Associated Steatotic Liver Disease (MASLD). They used HepG2 cells as an in vitro model. Overall, the authors provided a well-designed manuscript with a thorough methods section. However, the biological significance and rationale of the study need further elaboration, and the primary objectives require more detailed clarification.
The introduction section effectively presents major topics, but my major concern is that the study solely relied on the human hepatoblastoma HepG2 cell line. Including additional cell models would enhance the study's robustness. The authors should provide a justification for their choice of this particular cell model. Additionally, it is important to specify whether cell viability was assessed immediately or after 24 hours, and explain why hydrogen-rich water (HRW) studies were performed only for 24 hours.
Certain claims in the manuscript are overly assertive and need to be revised. For instance, the statement that "molecular hydrogen has a promising role in treating fatty liver and consistently demonstrates positive effects, particularly in reducing oxidative stress, which is the main factor causing fatty liver to progress to NASH" should be rephrased. It would be more appropriate to say, "Molecular hydrogen demonstrates positive effects, particularly in reducing oxidative stress, an important factor associated with the progression of fatty liver to NASH, suggesting its promising role in treating fatty liver disease." Additionally, the study's limitations, such as the reliance on a single cell model, should be discussed in the discussion section.
Author Response
Reviewer 1
In this study, the authors investigated the accumulation of intracellular lipid droplets in hepatocytes induced by free fatty acids to better understand Metabolic Dysfunction-Associated Steatotic Liver Disease (MASLD). They used HepG2 cells as an in vitro model. Overall, the authors provided a well-designed manuscript with a thorough methods section. However, the biological significance and rationale of the study need further elaboration, and the primary objectives require more detailed clarification.
Ans: Thank you for your positive remarks regarding the design and methods section of our manuscript. We agree that further elaboration on the biological significance and rationale of our study is necessary. We have now included additional context in the introduction section to better explain the importance of studying MASLD and the specific objectives of our research. This includes a detailed discussion on the relevance of intracellular lipid droplet accumulation in hepatocytes and how it contributes to our understanding of MASLD. (page 2, line 129-134)
The introduction section effectively presents major topics, but my major concern is that the study solely relied on the human hepatoblastoma HepG2 cell line. Including additional cell models would enhance the study's robustness. The authors should provide a justification for their choice of this particular cell model. Additionally, it is important to specify whether cell viability was assessed immediately or after 24 hours, and explain why hydrogen-rich water (HRW) studies were performed only for 24 hours.
Ans: We appreciate your concern regarding the use of a single cell model. The HepG2 cell line was chosen due to its widespread use in studying hepatic metabolism and its well-characterized response to FFA-induced lipid accumulation. We agree that including additional cell models would enhance the robustness of our study; however, we are unable to include experimental data from more cell lines because of time constraints; nonetheless, we intend to incorporate these models in future investigations.
Regarding cell viability, we have clarified in the methods section that cell viability was assessed for 24 hours to provide a comprehensive understanding of HRW's effects (page 3, line 584-585). The 24-hour timeframe for HRW studies was selected to capture the short-term effects of HRW on lipid accumulation and oxidative stress, based on previous studies suggesting significant changes within this period. This rationale has also been included in the revised manuscript. (page 6, line 939-941)
Certain claims in the manuscript are overly assertive and need to be revised. For instance, the statement that "molecular hydrogen has a promising role in treating fatty liver and consistently demonstrates positive effects, particularly in reducing oxidative stress, which is the main factor causing fatty liver to progress to NASH" should be rephrased. It would be more appropriate to say, "Molecular hydrogen demonstrates positive effects, particularly in reducing oxidative stress, an important factor associated with the progression of fatty liver to NASH, suggesting its promising role in treating fatty liver disease." Additionally, the study's limitations, such as the reliance on a single cell model, should be discussed in the discussion section.
Ans: We appreciate your guidance on improving the tone of our manuscript. We have revised the statement as suggested to provide a more balanced perspective: "Molecular hydrogen demonstrates positive effects, particularly in reducing oxidative stress, an important factor associated with the progression of fatty liver to NASH, suggesting its promising role in treating fatty liver disease." (page 12, line 1495-1497)
Additionally, we have included a section in the discussion that addresses the limitations of our study, particularly the reliance on a single cell model. This addition acknowledges the need for future studies to include multiple cell models to validate our findings and enhance the robustness of our conclusions. (page 13, line 1908-1913)
Thank you once again for your valuable feedback. We believe these revisions have strengthened our manuscript and look forward to any further comments you may have.

Reviewer 2 Report
Comments and Suggestions for Authors
1. The Abstract is too long (335 words).
2. LD is not defined at first mention. Fix.
3. Maintain a consistent tone. For example, in the Abstract, you present the results in the past tense and in the present tense. Stay consistent.
4. Line 139 - (Refs)?
5. Normal primary hepatocytes or cell lines (e.g., THLE-2) as opposed to tumor (HepG2) hepatocytes would have been a more appropriate experimental model for this study. Please explain.
6. Please add to the methods how the cells were checked for mycoplasma contamination.
7. Please add the concentrations of palmitic acid and oleic acid used for treatment. Also, since fatty acids are insoluble in water-based solvent and require organic solvents (e.g., DMSO, EtOH... etc.), the authors need to clarify in the methods how these two fatty acid were dissolved.
8. How long was the treatment? What was the vehicle? What was its concentration in percentage v/v? All of these details must be added to the methods.
9. Clarify the purpose of DMSO in the lipid accumulation assay.
10. "Chekine colorimetric commercial kit" - ChecKine kits since you are measuring TG and TC. Poor grammar.
11. Replace "ascertain" with "detect" or "measure" or "assay" or any suitable word. The English must be improved throughout.
12. For the lipid peroxidation measurement, did the authors use the kit as a colorimetric or a fluorimetric kit? Clarify.
13. Change qPCR to qRT-PCR. Also, did the authors design their own primers? If so, please mention the software used. Otherwise, the name of the manufacturer must be added.
14. In the Statistics section, why was the SEM used instead of the SD? Also, clearly specify how many independent experiments were performed and how many technical replicates were run in each.
15. Fig. 1E, change the x-axis label to "Lipid content (%)" or "Lipid accumulation (%)". Do this for all figures; label with what is actually being measured.
16. In Fig. 4A and 4B, there are comparison lines that do not show any symbol. Please fix.
Comments on the Quality of English LanguageThe language is not acceptable with considerable spelling mistakes, poor grammar and sentence structure, and redundancy. Have the manuscript revised by a science-literate native speaker or procure a professional editing service.
Author Response
Reviewer 2
We sincerely appreciate your careful review and thoughtful suggestions for our manuscript. Your feedback has been invaluable in improving the quality and clarity of our work. Below are our responses to your comments and concerns:
- The Abstract is too long (335 words).
Ans: We have rewritten the abstract and reduced it to 194 words.
- LD is not defined at first mention. Fix.
Ans: We have listed lipid droplets and their abbreviations (LDs) at the first occurrence in the text (abstract; page 2, line 129; and page 8, line 1114). In addition, we also gave a brief introduction to the properties and characteristics of LDs (page 2, line 129-131).
- Maintain a consistent tone. For example, in the Abstract, you present the results in the past tense and in the present tense. Stay consistent.
Ans: We have unified the tense throughout the text and thoroughly corrected all text content. Thank you very much for correcting our language deficiencies.
- Line 139 - (Refs)?
Ans: The content here was mistakenly posted and has been removed.
- Normal primary hepatocytes or cell lines (e.g., THLE-2) as opposed to tumor (HepG2) hepatocytes would have been a more appropriate experimental model for this study. Please explain.
Ans: Thank you for your insightful comment regarding the use of normal primary hepatocytes or cell lines such as THLE-2 as a more appropriate experimental model. We acknowledge that HepG2 cells, being cancerous, have limitations. However, they are widely used in similar research due to their well-characterized nature and consistent response to various treatments, making them a suitable model for initial investigations. We strongly agree with your suggestion to use normal liver cells for future experiments. Unfortunately, due to limited response time, we were unable to incorporate these models in the current study. We have addressed this limitation in the discussion section and plan to use normal primary hepatocytes or alternative cell lines in subsequent research to validate and expand upon our findings. Thank you once again for your valuable feedback.
- Please add to the methods how the cells were checked for mycoplasma contamination.
Ans: We used the Mycoplasma PCR Detection Kit from Applied Biological Materials Inc. (abm), catalog number G238, which is designed for the specific and sensitive detection of Mycoplasma contamination in cell cultures. To ensure the quality of our cell cultures, we conduct these tests every two months. Related descriptions have been added to Materials and Methods. (page 3, line 570-573)
- Please add the concentrations of palmitic acid and oleic acid used for treatment. Also, since fatty acids are insoluble in water-based solvent and require organic solvents (e.g., DMSO, EtOH... etc.), the authors need to clarify in the methods how these two fatty acid were dissolved.
Ans: Thank you for your insightful comments and suggestions. We have added the concentrations of palmitic acid and oleic acid used for the treatments in our revised methods section. Specifically, we used a final concentration of 1 mM for both palmitic acid and oleic acid. To address the solubility concerns, we prepared the fatty acid stock solutions as follows:
Palmitic Acid: We dissolved palmitic acid in a minimal amount of ethanol. To ensure complete dissolution, we gently heated the solution at 70°C with continuous stirring. Sodium hydroxide (0.1 M) was added dropwise as necessary. The final concentration of the stock solution was 100 mM.
Oleic Acid: Similarly, oleic acid was dissolved in ethanol and heated gently at 70°C with continuous stirring. Sodium hydroxide (0.1 M) was also added dropwise to facilitate complete dissolution. The final concentration of this stock solution was 100 mM.
Both fatty acids were then conjugated with a 10% BSA solution, prepared in distilled water and adjusted to a pH of approximately 7.4. The fatty acids were mixed in a 1:2 ratio (PA:OA) and added to the BSA solution at a molar ratio of 6:1 (fatty acid to BSA). The mixture was thoroughly stirred and incubated at 37°C for 12 hour to ensure proper binding. After conjugation, adjust the final volume to achieve the desired concentration. The final conjugated fatty acid-BSA solution was filter-sterilized and stored at -20°C for long-term use. Related descriptions have been added to Materials and Methods (page 3, line 575-583). We hope these clarifications address your concerns. Thank you once again for your valuable feedback.
- How long was the treatment? What was the vehicle? What was its concentration in percentage v/v? All of these details must be added to the methods.
Ans: Thank you for your comments and the opportunity to clarify our methods. Here are the additional details you requested:
Treatment Duration: The treatment with FFA was conducted for 24 hours.
Vehicle: The vehicle used was a BSA solution without FFA.
Concentration: The final concentration of the FFA (conjugated fatty acid-BSA solution) was 1 mM.
These details have been added to the methods section to ensure clarity and completeness.
- Clarify the purpose of DMSO in the lipid accumulation assay.
Ans: Thank you for your question regarding the purpose of DMSO in the lipid accumulation assay. In our study, DMSO was primarily used to dissolve the Oil Red-O dye from the cells after staining. This step is essential for subsequent spectrophotometric quantification, which measures the lipid content in the cells. By dissolving the dye, DMSO ensures accurate and reliable quantification of lipid accumulation. (page 4, line 674-676)
We hope this clarification addresses your query. Thank you for your valuable feedback.
- "Chekine colorimetric commercial kit" - ChecKine kits since you are measuring TG and TC. Poor grammar.
Ans: Thank you for your feedback. We have corrected the grammar in the relevant section (page 4, line 679-681). Additionally, the relevant product links for the CheKine colorimetric commercial kits, which we used to measure triglycerides (TG) and total cholesterol (TC), are provided below:
TG: https://www.abbkine.com/product/chekine-micro-total-cholesterol-tc-assay-kit-ktb2220/
TC: https://www.abbkine.com/product/chekine-micro-triglyceride-tg-assay-kit-ktb2200/
We appreciate your suggestions and believe the revisions improve the clarity and completeness of our methods section.
- Replace "ascertain" with "detect" or "measure" or "assay" or any suitable word. The English must be improved throughout.
Ans: Thank you for your feedback. We have replaced the word "ascertain" with more appropriate terms such as "detect," "measure," or "assay" throughout the manuscript. Additionally, we have corrected the grammar and improved the overall English of the entire text to enhance clarity and readability.
- For the lipid peroxidation measurement, did the authors use the kit as a colorimetric or a fluorimetric kit? Clarify.
Ans: Thank you for your query regarding the lipid peroxidation measurement. We used the BioVision Lipid Peroxidation (MDA) Assay Kit in its colorimetric mode (OD = 532 nm). Additionally, BioVision has been officially acquired by Abcam in 2021. Therefore, the detailed name of the kit should refer to Abcam. You can find more information on the kit here: https://www.abcam.com/products/assay-kits/lipid-peroxidation-mda-assay-kit-colorimetricfluorometric-ab118970.html
The original name of the reagent specified by the manufacturer was "Lipid Peroxidation Colorimetric/Fluorometric Assay Kit", but in order to prevent confusion, we changed it to "Lipid Peroxidation Assay Kit", and noted that we are using colorimetric mode (OD = 532 nm). (page 4, line 711)
- Change qPCR to qRT-PCR. Also, did the authors design their own primers? If so, please mention the software used. Otherwise, the name of the manufacturer must be added.
Ans: Thank you for your comments and suggestions. We have updated the manuscript to change "qPCR" to "qRT-PCR" to accurately reflect the technique used. Regarding the primers, we designed our own primers using ABI's Primer Express software. This detail has been added to the methods section for clarity. (page 5, line 836-837)
- In the Statistics section, why was the SEM used instead of the SD? Also, clearly specify how many independent experiments were performed and how many technical replicates were run in each.
Ans: Thank you for your questions regarding the statistical analysis. We used the standard error of the mean (SEM) instead of the standard deviation (SD) to provide a measure of the precision of the sample mean, which is particularly useful when comparing the means of multiple groups. We chose to use the SEM instead of the SD in order to demonstrate whether the obtained mean can reliably reflect the overall population mean. This choice underscores the performance of mean stability by indicating the precision of the sample mean as an estimate of the population mean. Using SEM provides a clearer understanding of the reliability and accuracy of the mean values derived from our experiments.
We conducted all relevant experiments in independent triplicates. Each independent experiment included three technical replicates to ensure the reliability and reproducibility of our results. (page 5, line 845-846)
- Fig. 1E, change the x-axis label to "Lipid content (%)" or "Lipid accumulation (%)". Do this for all figures; label with what is actually being measured.
Ans: Thank you for your feedback and suggestions regarding the figure labels. We have made the requested corrections in Fig. 1E, changing the x-axis label to "Lipid content (%)" to accurately reflect what is being measured. Additionally, we have updated Fig. 3B to include "% of control" to provide a more realistic representation of the data. We appreciate your guidance in improving the clarity and precision of our figures.
- In Fig. 4A and 4B, there are comparison lines that do not show any symbol. Please fix.
Ans: Thank you for your observations regarding the comparison lines in Fig. 4A and 4B. We have removed the inappropriate comparison lines and ensured that the relevant post-comparison pairs are clearly marked in the figure legends. We appreciate your feedback, which has helped us improve the clarity and accuracy of our figures.

Round 2
Reviewer 2 Report
Comments and Suggestions for Authors
The authors have adequately addressed the comments.
Comments on the Quality of English LanguageMinor issues.